# The study on setting priorities of zoonotic agents for medical preparedness and allocation of research resources

**Kung-Ching Wang** [1], **Chia-Lin Chang**[2], **Sung-Hsi Wei**[3], **Chao-Chin Chang** [1]*

1 Graduate Institute of Microbiology and Public Health, National Chung Hsing University, Taichung, Taiwan, R.O.C, 2 Department of Applied Economics, National Chung Hsing University, Taichung, Taiwan, R.O.C, 3 Children's Hospital, China Medical University, Taichung, Taiwan, R.O.C

* changcc@dragon.nchu.edu.tw

**Data Availability Statement:** All data relevant to the analysis in this study were collected from the websites or references described in the Materials and Methods. For transformed data, all information was summarized in Tables 2, 4, 5, 6, and 9.

## Abstract

The aim of this study is to develop a scoring platform to be used as a reference for both medical preparedness and research resource allocation in the prioritization of zoonoses. Using a case-control design, a comprehensive analysis of 46 zoonoses was conducted to identify factors influencing disease prioritization. This analysis provides a basis for constructing models and calculating prioritization scores for different diseases. The case group (n = 23) includes diseases that require immediate notification to health authorities within 24 hours of diagnosis. The control group (n = 23) includes diseases that do not require such immediate notification. Two different models were developed for primary disease prioritization: one model incorporated the four most commonly used prioritization criteria identified through an extensive literature review. The second model used the results of multiple logistic regression analysis to identify significant factors (with p-value less than 0.1) associated with 24-hour reporting, allowing for objective determination of disease prioritization criteria. These different modeling approaches may result in different weights and positive or negative effects of relevant factors within each model. Our study results highlight the variability of zoonotic disease information across time and geographic regions. It provides an objective platform to rank zoonoses and highlights the critical need for regular updates in the prioritization process to ensure timely preparedness. This study successfully established an objective framework for assessing the importance of zoonotic diseases. From a government perspective, it advocates applying principles that consider disease characteristics and medical resource preparedness in prioritization. The results of this study also emphasize the need for dynamic prioritization to effectively improve preparedness to prevent and control disease.

## 1. Introduction

Zoonoses refers to diseases that that are naturally transmitted between vertebrate animals and man," as defined in 1951 by the World Health Organization (WHO) Expert Committee on Zoonoses [1]. Approximately 60% of known human infectious diseases and 75% of emerging infectious diseases are caused by zoonotic agents [2]. With the expansion of urbanization and

**Funding:** The study was partially supported by the National Science and Technology Council, Executive Yuan, Taiwan in the form of a grant to C-CC [NSTC-111-2313-B-005-045-MY2].

**Competing interests:** The authors have declared that no competing interests exist.

agriculture, humans are coming into more frequent contact with wildlife. Additionally, climate change and global trade also contribute to the spread of zoonses. International travel and population movement facilitate the global spread of infectious diseases. Once a zoonosis has been introduced into a country, factors such as urbanization and an aging population increase the likelihood of disease transmission and cause high case-fatality rates in humans. Moreover, several factors related to the characteristics of the pathogen, including its mode of transmission, influence the magnitude of an epidemic of a zoonosis. The availability of therapeutic agents or vaccines to prevent viral or bacterial diseases also determines the epidemic scale of a zoonosis once occurred. As a result, preventing and controlling zoonoses have become critical public health issues worldwide [3].

In recent years, the discovery of more emerging and re-emerging zoonoses has prompted countries to develop effective response measures and prepare relevant medical resources. However, regarding the purpose for development strategies for disease prevention and diagnosis, it is also necessary to establish a priority ranking system for communicable diseases to allocate research resources. Up to date, the methodology of One Health Zoonotic Disease Prioritization (OHZDP) has been recommended by CDC in the US as a tool for zoonoses prioritization, and various methods used for this purpose in different countries include the Hirsch index (h-index), Delphi technique, multi-criteria decision analysis (MCDA), and questionnaires; each method is with its advantages and disadvantages [4]. Consequently, achieving a consensus on the methods for prioritizing diseases is challenging [5].

Therefore, the purpose of this study is to respectively establish medical preparedness priority ranking and research priority ranking systems for zoonotic infectious diseases. Using a case-control study for comparing relative importance of zoonoses, statistical analyses on the corresponding epidemiological data were conducted to identify the most critical factors for setting priorities. Subsequently, we use the constructed statistical model to calculate the ranking score for each zoonoses, which can be applied for disease prioritization.

## 2. Materials and methods

### Developing a disease priority ranking method with binary logistic regression model

This study involved two main models: Models A and B. The factors used in Model A were chosen through literature review approach, and the frequently used four factors for disease prioritization were included in the model after literature summarization (please refer to description given below). For the construction of Model B, the influencing factors used in this study for prioritizing diseases were based on results of statistical analysis using a case-control study design approach; categorical variables were analyzed using Chi-square test or Fisher's exact test, and continuous variables were analyzed using independent t-test. Based on the results of the univariate analysis, a binary logistic regression model was used for constructing the multiple logistic regression model to determine the weight of each factor for disease prioritization. While statistical significance of a factor was determined with $p < 0.05$, factors with $p < 0.1$ in the univariate analysis were still included in the multiple logistic model to adjust potential confounding effect.

The dependent variable in the binary logistic regression model was based on the concept of a case-control study, with diseases requiring reporting within 24 hours considered as the case group and diseases not requiring such reporting as the control group. After completing the construction of the model, the weight for each factor was determined based on the obtained odds ratio (OR) value: OR $\geq$ 4 or $\leq$ 0.25 received a weight of 4; OR values between 3–4 or 0.25–0.33 received a weight of 3; OR values between 2–3 or 0.33–0.5 received a weight of 2, and OR values between 1–2 or 0.5–1 received a weight of 1. In Models A and B, further sub-

models (Models A.1, A.2, and B.1, B.2) were constructed based on the different prioritization needs, namely "the characteristics of the disease itself and the ability to prepare medical resources" or "the need for stricter border controls and enhanced research on vaccine development or therapeutic drugs", as these two purposes may lead to totally different prioritization ranking system while assigning a positive or negative value of the weight.

Finally, after the weight of a factor has been determined, the total score for each disease was calculated for prioritization using the formulated overall equation. Data analysis was performed using IBM SPSS Statistics for Windows, Version 20.0. Armonk, NY: IBM Corp.

## Criteria of literature search and review for the study

Using "priorit* and zoono*/disease*" as keywords, a literature search was conducted in PubMed for articles published between 2010 and 2020, resulting in a total of 713 articles. After further screening the titles to remove misclassified articles, 55 relevant articles remained. These 55 articles were further filtered based on the following criteria: published in English, containing methods for prioritizing diseases, listing prioritization criteria, and with substantial results. This resulted in a final selection of 25 relevant articles for literature summarization in this study (Table 1) [6–30]. Through careful review of the 25 relevant articles, frequency of the

**Table 1. Prioritization methods employed in the relevant 25 studies [6–30].**

| Author (year) | Prioritization Methods |
|---|---|
| Doherty, J.A. et al (2006) [6] | Delphi technique |
| Krause, G. et al (2008) [7] | Delphi panel |
| Cardoen, S. et al (2009) [8] | Working group |
| Havelaar, A.H. et al (2010) [9] | Multicriteria analysis |
| Balabanova, Y. et al (2011) [10] | Delphi panel |
| Cox, R. et al (2012) [11] | Delphi-like approach |
| Humblet, M.F. et al (2012) [12] | 1.Deterministic with mean of weight<br>2. Functions of weights by using Monte Carlo simulation |
| Cediel, N. et al (2013) [13] | Delphi panel |
| Cox, R. et al (2013) [14] | Multicriteria decision analysis |
| Economopoulou, A. et al (2014) [15] | Consensus-building Delphi method and a risk matrix |
| Kurain, A. et al (2014) [16] | Composite Index |
| Dahl, V. et al (2015) [17] | Delphi process |
| Kadohira, M. et al (2015) [18] | Analytic hierarchy process method |
| Munyua, P. et al (2015) [19] | OHDZP tool |
| Stebler, N. et al (2015) [20] | Modified Delphi panel |
| Trang do, T. et al (2015) [21] | Questionnaire |
| Hongoh, V. et al (2016) [22] | Multicriteria decision analysis |
| McFadden, A.M. et al (2016) [23] | Multicriteria ranking modelling |
| Stebler, N. et al (2016) [24] | Conjoint analysis questionnaire |
| Mehand M.S. et al (2018) [25] | Delphi technique, questionnaires,<br>multicriteria decision analysis, and expert review |
| Mehand M.S. et al (2018) [26] | Delphi technique, questionnaires, multicriteria decision analysis, and expert review |
| Sekamatte, M. et al (2018) [27] | OHDZP tool |
| Yasobant, S. et al (2019) [28] | OHDZP tool |
| Zecconi, A. et al (2019) [29] | Discontools based on scorecards |
| Klamer, S. (2021) [30] | Multicriteria decision analysis |

17 criteria were then summarized, and the top 3 criteria were identified and further used for zoonosis prioritization in this study.

**Disease selection for the study.** Notifiable diseases refer to communicable diseases classified by Taiwan's Centers for Disease Control and Prevention (CDC) based on the level of risk, such as mortality rate, incidence rate, and transmission speed; cases related to all notifiable diseases must be reported to the CDC. As the main focus of this study is related to prioritization of "zoonoses", the diseases were further selected from the CDC's website. Furthermore, a case-control study was applied for comparison to determine the most influential factors associated with disease prioritization through statistical analysis. The cases included a total of 23 diseases that require reporting within 24 hours, while the controls included also 23 diseases that do not require such reporting (Table 2).

## Data collection for factors relevant to disease prioritization

The case-fatality rate in humans, availability of treatment and vaccines for humans, pathogenicity, and transmission modes may vary due to new study information and different data resources. This study therefore collected relevant data from international references such as

**Table 2. The list of zoonoses used for disease prioritization in this study.**

| Notifiable zoonoses within 24 hours[a] (n = 23) | Zoonoses not required to be reported within 24 hours[b] (n = 23) |
|---|---|
| Rabies | Japanese encephalitis |
| Plague | Listeriosis |
| SARS[c] | Scrub typhus |
| Dengue fever | Endemic typhus |
| Chikungunya fever | Lyme disease |
| Zika viral infection | New variant Creutzfeldt-Jakob disease |
| West Nile fever | Toxoplasmosis |
| Epidemic typhus | Brucellosis |
| Enteropathogenic *Escherichia coli* | Q fever |
| Anthrax | Tularemia |
| Hantavirus syndrome | Bovine tuberculosis |
| SFTS[d] | Salmonellosis |
| Melioidosis | *Angiostrongylus cantonensis* infection |
| Leptospirosis | *Paragonimus* infection |
| Herpesvirus B infection | Trichinosis |
| Novel influenza A virus infection | Blastocystosis |
| Yellow fever | Cryptosporidiosis |
| Rift Valley fever | Psittacosis |
| MERS-CoV[e] infection | Hendra viral infection |
| Lassa fever | Cat-scratch disease |
| Marburg viral hemorrhagic fever | *Clonorchis sinensis* infection |
| Ebola viral infection | *Streptococcus suis* type 2 infection |
| Severe COVID-19 infection | Nipah viral infection |

[a]Defined as cases in the case-control study.

[b]Defined as controls in the case-control study.

[c]SARS: Severe acute respiratory syndrome.

[d]SFTS: Sever fever with thrombocytopenia syndrome.

[e]MERS-CoV: Middle East respiratory syndrome coronavirus.

"Mandell, Douglas, and Bennett's Principles and Practice of Infectious Diseases" [31], as well as the websites of the Centers for Disease Control and Prevention (CDC) of the United States [32], and the World Health Organization (WHO) [33], to gather these information. Regarding the disease incidence, we used the number of confirmed disease cases from 2018 to 2020 obtained from the website of the Taiwan Centers for Disease Control and Prevention (CDC) infectious disease statistics query system [34], and the mid-year population of Taiwan from 2018 to 2020 obtained from the website of the Ministry of the Interior [35] to calculate yearly incidence for notifiable diseases.

For non-notifiable diseases, incidence data was collected through literature review from PubMed using the disease name and "Taiwan" as keywords. To identify whether the outbreak has been ever occurred in Taiwan and any arthropod vector responsible for the transmission, we used the disease name and "Taiwan" as keywords to search for relevant literature in PubMed and the epidemic report from Taiwan CDC. Regarding the case-fatality rate and availability of vaccines in animals, as well as the animal species that could be infected, the information was collected from the official websites of the Iowa State University Food Safety and Public Health Center [36] (The Center for Food Security and Public Health, Iowa State University, USA 2022), the World Organization for Animal Health (WOAH) [37], and the MSD Veterinary Manual website [38]. If no data was available from these sources, relevant literature was searched using the disease's English name in PubMed. The statistics annual reports published by the Bureau of Animal and Plant Health Inspection and Quarantine (BAPHIQ) [39], Taiwan, from 2018 to 2020 were reviewed to determine whether the disease has ever occurred in animals. Whether the pathogen can be used as bioterrorism agent was checked in the website of the Centers for Disease Control and Prevention in USA [40]. Relevant information collected from World Health Organization (WHO) [41,42] were collected to identify whether the disease occurs in humans or animals needs to be reported to WHO or WOAH, respectively.

## Definition of the risk score of a country associated with Taiwan

To determine the risk score of a country, the following steps were taken and analyzed. Disease occurrence data from 2010 to 2020 were collected from the websites of National Health Commission in People's Republic of China [43], National Institute of Infectious Diseases in Japan [44], Disease Management Headquarters in South Korea [45], Epidemiology Bureau, Department of Health in the Philippines [46], Ministry of Health Portal in Vietnam [47], Department of Disease Control in Thailand [48], Ministry of Health in Indonesia [49], Ministry of Health in Malaysia [50], Ministry of Health in Singapore [51], Centre for Health Protection in Hong Kong [52], and the Centers for Disease Control and Prevention in USA [53]. If a country had no relevant data on a particular disease, PubMed was used to search for related literature using the disease name and the country's name. Furthermore, the risk assessment of countries closely related to Taiwan was based on data collected from the Tourism Bureau's tourism statistics database of inbound travelers to Taiwan from 2010 to 2020 [54]. The data on the number of residents in each country and the number of Taiwanese outbound travelers to each country were combined. The top 11 countries with the highest total number of inbound travelers, outbound travelers, and migrant workers were selected for further evaluation; a risk score (from 1 to 4) was assigned to each country based on the quartile distribution of the total number of people and whether the disease has ever occurred in the country from 2010 to 2020 [55].

**The definition of a score to present the importance of diversity of transmission routes of a disease.** Different zoonoses may have various transmission routes. The more diverse these routes are, the more challenging on disease control becomes. Therefore, the weights of

transmission routes were determined by assessing five variables: contact transmission, airborne transmission, foodborne transmission, vector-borne transmission, and person-to-person transmission. These variables were incorporated into the multiple logistic regression model after a case-control study analysis as defined above, and corresponding scores were assigned based on the calculated odds ratio (OR) values. Based on the OR values after such statistical analysis, a score of 4 was assigned for foodborne or person-to-person transmission, 3 for contact or vector-borne transmission, and 2 for airborne transmission. As a disease may have various transmission routes, the total score for each disease based on their demonstrated transmission routes was calculated. The scores of all diseases evaluated in this study were then subjected to distribution analysis and score quartiles were determined. Using the quartiles, a weighted value of 4 of a disease was assigned if the total score was $\geq 9$, $\geq 4$ and $< 9$ received a weighted value of 3, a score of 3 received a weighted value of 2, and the score less than 3 received a weighted value of 1. The weighted value was finally used to present diversity of transmission routes of a disease.

## 3. Results

### Identification of factors for model construction and settings of their weights

In Model A, 25 relevant literatures were reviewed. The results showed that the three most commonly used conditions, were "severity of the disease in humans", "occurrence of the disease in the country", and "curability in humans" (Table 3).

The factor "availability of prevention measures in humans" and the factor "economic losses" were tied in the 4th rank for common usage. However, the economic loss factor was not included in our final model, since its definition was relevant to whether the infection in

**Table 3. The summarized results of 25 pertinent articles on disease prioritization study.**

| Rank | Applied factor in an article | The factor name used in our study | The frequency used in articles |
|---|---|---|---|
| 1 | Severity in humans | Case-fatality rate in humans over 5% | 22 |
| 2 | The disease ever occurred in a country in humans | The disease ever identified in humans in Taiwan, 2018–2020 | 18 |
| 3 | With therapeutic drugs | With available antibiotics, anti-viral drugs or antibodies for therapy in humans | 13 |
| 4 | With preventive measures in humans | Vaccine available in humans | 10 |
| 4 | Economic loss | The disease required to be reported to WOAH | 10 |
| 6 | The disease ever occurred in a country in animals | The disease ever identified in animals in Taiwan, 2018–2020 | 9 |
| 6 | Severity in animals | Case-fatality rate in animals over 5% | 9 |
| 8 | Mode of transmission | Any mode of transmission through inhalation, fecal oral, contact, arthropod, or human-to-human transmission | 8 |
| 9 | International responsibility | The disease required to be reported to WHO | 6 |
| 10 | Any outbreaks occurred in a country | Any identified outbreaks in Taiwan in the past 20 years | 4 |
| 10 | Impact on international trade | Quarantine required in imported swine, ruminant and avian species | 4 |
| 12 | With preventive measures in animals | Vaccine available in animals | 3 |
| 13 | The scale of affected economic animal species | The disease can infect swine, ruminant or avian species | 3 |
| 14 | Type of pathogens | Viruses, bacteria, parasites, fungi or others | 3 |
| 15 | Any identified arthropod species in a country | Any identified arthropod species in Taiwan | 2 |
| 16 | Bioterrorism agents | Listed as a bioweapon in CDC, USA | 2 |
| 17 | The likelihood of disease introduction from other countries | Risk score of countries associated with Taiwan | 2 |

animals needed to be reported to the World Organization for Animal Health (WOAH), which was based on disease and pathogen characteristics. Therefore, it could potentially lead to collinearity issues with the other factors in the model. Consequently, the four most commonly used conditions corresponded to the factors "human case fatality rate exceeding 5%," "occurrence in Taiwan," "curability in humans," and "availability of prevention measures in humans" were finally included in Model A.

In Model B, regarding the impact based on the risk score of a country associated with Taiwan, the total impact score and weight for each disease were calculated (Table 4). Diseases that received a weight score of 4 included dengue fever, Chikungunya fever, Zika viral infection, melioidosis, leptospirosis, severe COVID-19 infection, Japanese encephalitis, listeriosis, scrub typhus, toxoplasmosis, Q fever, salmonellosis, cryptosporidiosis, and *Streptococcus suis* type 2 infection. These diseases with easily transmissible nature received high scores primarily due to their wide-ranging impact on multiple countries.

Regarding the impact of multiple transmission routes for each disease, the results showed that diseases receiving a weight score of 4 for the transmission route included plague, anthrax, melioidosis, novel influenza A viral infection, MERS-CoV2 viral infection, Lassa fever, brucellosis, Q fever, tularemia, bovine tuberculosis, salmonellosis, and Nipah viral infection (please refer to Table 5).

**The overall prioritization results after the analysis of Model A.** As the above results, Model A utilizes four factors: "case-fatality rate in humans > 5%," "the disease ever occurred

**Table 4. The weight of a disease reflects a country's risk score in relation to Taiwan.**

| Disease | Score[a] | Weight | Disease | Score | Weight |
|---|---|---|---|---|---|
| Rabies | 18 | 3 | Japanese encephalitis | 22 | 4 |
| Plague | 7 | 2 | Listeriosis | 22 | 4 |
| SARS | 0 | 0 | Scrub typhus | 21 | 3 |
| Dengue fever | 27 | 4 | Endemic typhus | 20 | 3 |
| Chikungunya fever | 27 | 4 | Lyme disease | 14 | 2 |
| Zika viral infection | 23 | 4 | New variant Creutzfeldt-Jakob disease | 0 | 0 |
| West Nile fever | 12 | 2 | Toxoplasmosis | 23 | 4 |
| Epidemic typhus | 0 | 0 | Brucellosis | 18 | 3 |
| Enteropathogenic *Escherichia coli* | 21 | 3 | Q fever | 22 | 4 |
| Anthrax | 13 | 2 | Tularemia | 11 | 2 |
| Hantavirus syndrome | 14 | 2 | Bovine tuberculosis | 10 | 2 |
| SFTS | 15 | 2 | Salmonellosis | 27 | 4 |
| Melioidosis | 27 | 4 | *Angiostrongylus cantonensis* infection | 12 | 2 |
| Leptospirosis | 25 | 4 | *Paragonimus* infection | 18 | 3 |
| Herpesvirus B infection | 4 | 1 | Trichinosis | 16 | 3 |
| Novel influenza A virus infection | 11 | 2 | Blastocystosis | 17 | 3 |
| Yellow fever | 0 | 0 | Cryptosporidiosis | 27 | 4 |
| Rift Valley fever | 0 | 0 | Psittacosis | 20 | 3 |
| MERS-CoV infection | 3 | 1 | Hendra viral infection | 0 | 0 |
| Lassa fever | 0 | 0 | Cat-scratch disease | 17 | 3 |
| Marburg viral hemorrhagic fever | 0 | 0 | *Clonorchis sinensis* infection | 13 | 2 |
| Ebola viral infection | 4 | 1 | *Streptococcus suis* type 2 infection | 25 | 4 |
| Severe COVID-19 infection | 27 | 4 | Nipah viral infection | 0 | 0 |

[a] The score presented the sum of the risk from various countries, and was calculated based on the disease ever occurred in China, Japan and Hong Kong (score = 4), in South Korea and USA (score = 3), in Thailand, Viet Nam and Singapore (score = 2) and in Malaysia, Philippines and Indonesia (score = 1).

**Table 5. The impact of score calculation and the weight for prioritization use in diseases with multiple routes of transmission.**

| Disease | C/I/FO/A/HH[a] | Score[b] | Weight | Disease | C/I/FO/A/HH[a] | Score[b] | Weight |
|---------|----------------|----------|--------|---------|----------------|----------|--------|
| Rabies | Y/Y/N/N/N | 5 | 3 | Japanese encephalitis | N/N/N/Y/N | 3 | 2 |
| Plague | Y/Y/N/Y/Y | 12 | 4 | Listeriosis | N/N/Y/N/N | 4 | 3 |
| SARS | N/Y/N/N/Y | 6 | 3 | New variant Creutzfeldt-Jakob disease | N/N/Y/N/N | 4 | 3 |
| Dengue fever | N/N/N/Y/N | 3 | 2 | Endemic typhus | N/N/N/Y/N | 3 | 2 |
| Chikungunya fever | N/N/N/Y/N | 3 | 2 | Lyme disease | N/N/N/Y/N | 3 | 2 |
| Zika viral infection | N/N/N/Y/Y | 7 | 3 | Scrub typhus | N/N/N/Y/N | 3 | 2 |
| West Nile fever | N/N/N/Y/N | 3 | 2 | Toxoplasmosis | N/N/Y/N/N | 4 | 3 |
| Epidemic typhus | N/N/N/Y/N | 3 | 2 | Brucellosis | Y/Y/Y/N/N | 10 | 4 |
| Enteropathogenic *Escherichia coli* | N/N/Y/N/Y | 8 | 3 | Q fever | Y/Y/Y/N/N | 10 | 4 |
| Anthrax | Y/Y/Y/N/N | 9 | 4 | Tularemia | Y/Y/Y/Y/N | 13 | 4 |
| Hantavirus syndrome | Y/N/N/N/N | 3 | 2 | Bovine tuberculosis | Y/Y/Y/N/Y | 14 | 4 |
| SFTS | N/N/N/Y/N | 3 | 2 | Salmonellosis | Y/N/Y/N/Y | 12 | 4 |
| Melioidosis | Y/Y/Y/N/N | 9 | 4 | *Angiostrongylus cantonensis* infection | N/N/Y/N/N | 4 | 3 |
| Leptospirosis | Y/N/N/N/N | 3 | 2 | *Paragonimus* infection | N/N/Y/N/N | 4 | 3 |
| Herpesvirus B infection | Y/N/N/N/N | 3 | 2 | Trichinosis | N/N/Y/N/N | 4 | 3 |
| Novel influenza A virus infection | Y/Y/N/N/Y | 9 | 4 | Blastocystosis | N/N/Y/N/Y | 8 | 3 |
| Yellow fever | N/N/N/Y/N | 3 | 2 | Cryptosporidiosis | N/Y/N/N/N | 2 | 1 |
| Rift Valley fever | Y/N/N/Y/N | 6 | 3 | Psittacosis | N/Y/N/N/N | 2 | 1 |
| MERS-CoV infection | Y/Y/N/N/Y | 9 | 4 | Hendra viral infection | Y/N/N/N/N | 4 | 3 |
| Lassa fever | Y/Y/Y/N/Y | 13 | 4 | Cat-scratch disease | Y/N/N/N/N | 4 | 3 |
| Marburg viral hemorrhagic fever | Y/N/N/N/Y | 7 | 3 | *Clonorchis sinensis* infection | N/N/Y/N/N | 4 | 3 |
| Ebola viral infection | Y/N/N/N/Y | 7 | 3 | *Streptococcus suis* type 2 infection | Y/N/Y/N/N | 8 | 3 |
| Severe COVID-19 infection | N/Y/N/N/Y | 6 | 3 | Nipah viral infection | Y/N/Y/N/Y | 12 | 4 |

Y: With the specific route; N: Without the specific route.

[a] C: Contact, I: Inhalation, FO: Fecal-oral, A: Arthropod-borne, HH: Human-to-human.

in Taiwan," "the disease with therapeutic drugs", and "availability of prevention measures in humans" for multiple logistic regression analysis. After the regression analysis and based on the OR value derived from each factor, the weight was determined. As mentioning in the materials and methods, model construction needs to consider different purposes for disease prioritization. Therefore, further sub-models (Models A.1 and A.2) were constructed based on different prioritization needs, namely "the characteristics of the disease itself and the ability to prepare medical resources" or "the need for stricter border controls and enhanced research on vaccine development or therapeutic drugs". According to these two different main purposes, in Models A.1 and A.2, further consideration is subjectively to assign the positive or negative value for the weight of each factor for calculation of prioritization scores (Table 6).

Model A.1 is constructed based on the consideration of "disease characteristics and the availability of medical resources". The formula for calculating each disease priority ranking

**Table 6. The comparison of the variables and their weights used in calculation of the prioritization scores in models A.1 and A.2.**

| Variables | Notifiable zoonoses within 24 hours (n = 23) | Zoonoses not required to be reported within 24 hours (n = 23) | Odds ratio | The weight value used for score calculation[a] | The weight impact in Model A.1[b] | The weight impact in Model A.2[b] |
|---|---|---|---|---|---|---|
| Case-fatality rate over 5% in humans | 20 (87%) | 13 (57%) | 3.72 | 3 | + | + |
| The disease ever occurred in a country | 9 (39%) | 13 (57%) | 0.67 | 1 | + | + |
| The disease with therapeutic drugs | 12 (52%) | 19 (83%) | 0.25 | 4 | + | - |
| With available preventive measures | 10 (44%) | 4 (17%) | 4.50 | 4 | + | - |

[a]The weight for each factor was determined based on the obtained odds ratio (OR) value: OR $\geq$ 4 or $\leq$ 0.25 received a weight of 4; OR values between 3–4 or 0.25–0.33 received a weight of 3; OR values between 2–3 or 0.33–0.5 received a weight of 2, and OR values between 1–2 or 0.5–1 received a weight of 1.

[b]Models A.1 and A.2, were respectively constructed based on the different prioritization needs, namely "the characteristics of the disease itself and the ability to prepare medical resources" or "the need for stricter border controls and enhanced research on vaccine development or therapeutic drugs".

score is as follows: Total score = 3 * (case-fatality rate in humans > 5%) + 1 * (the disease ever occurred in Taiwan) + 4 * (the disease with therapeutic drugs in humans) + 4 * (with available preventive measures in humans). The results show that the diseases with the highest ranking are leptospirosis, bovine tuberculosis, and Hantavirus syndrome, all scoring 12 points. The diseases with the second-highest ranking are plague, anthrax, novel influenza A viral infection, Ebola viral infection, and tularemia, all scoring 11 points. Severe COVID-19 infection and Q fever ranked third, both scoring 9 points (see Table 7).

Model A.2 is constructed based on the consideration of "the need for stricter border controls and enhanced research on vaccines or therapeutic drugs for the disease". In comparison to Model A.1, the weights for "the disease with therapeutic drugs in humans" and "with available preventive measures in humans" are unchanged but subjectively assigned to negative values. The formula for calculating the disease priority ranking score is as follows: Total score = 3 * (case-fatality rate > 5% in humans) + 1 * (the disease ever occurred in Taiwan) - 4 * (the disease with therapeutic drugs in humans) - 4 * (with available preventive measures in humans). The results show that the diseases with the highest ranking are Zika viral infection and SFTS, both scoring 4 points. The diseases with the second-highest ranking are SARS, West Nile fever, Rift Valley fever, MERS-CoV2 viral infection, Marburg viral infection, Hendra viral infection, new variant Creutzfeldt-Jakob disease, and Nipah viral infection, all scoring 3 points. Chikungunya fever ranks third, scoring 1 point (Table 7).

**The overall prioritization results after the analysis of Model B.** Based on the case-control study design and the univariate analysis results (Table 8), a total of nine factors were found to meet the criteria for inclusion in the multiple logistic regression model with factors that showed p value less than 0.1. The factors include "human case-fatality rate of the disease >5%," "human case ever occurred in Taiwan, 2018 to 2020," "diverse transmission modes of the disease," "the disease with therapeutic drugs", "preventive measures available in humans," "pathogen type," "the disease can infect economic animals," "the disease needs to be reported to WHO," and "the risk score of a country associated with Taiwan". However, upon further consideration, the factor "disease needs to be reported to WHO" was excluded from the model, because it is a result based on WHO's consideration of disease characteristics, and it may exhibit collinearity with other related factors in the model (e.g., "human case fatality rate >5%"). Moreover, as "pathogen type" is highly correlated with "human case fatality rate >5%" (e.g., viral infections), it was also excluded from the model to avoid potential collinearity. The

**Table 7. Comparing prioritization results and disease scores from models A.1 and A.2[a].**

| Sub-model A.1 | The score | Sub-model A.2 | The score |
|---|---|---|---|
| Leptospirosis | 12 | Zika viral infection | 4 |
| Bovine tuberculosis | 12 | SFTS | 4 |
| Hantavirus syndrome | 12 | SARS | 3 |
| Plague | 11 | West Nile fever | 3 |
| Anthrax | 11 | MERS-CoV infection | 3 |
| Novel influenza A virus infection | 11 | Rift Valley fever | 3 |
| Ebola viral infection | 11 | Marburg viral hemorrhagic fever | 3 |
| Tularemia | 11 | Hendra viral infection | 3 |
| Severe COVID-19 infection | 9 | New variant Creutzfeldt-Jakob disease | 3 |
| Q fever | 9 | Nipah viral infection | 3 |
| Enteropathogenic *Escherichia coli* | 8 | Chikungunya fever | 1 |
| Melioidosis | 8 | Enteropathogenic *Escherichia coli* | 0 |
| Japanese encephalitis | 8 | Melioidosis | 0 |
| Listeriosis | 8 | *Angiostrongylus cantonensis* infection | 0 |
| Scrub typhus | 8 | Listeriosis | 0 |
| Salmonellosis | 8 | Scrub typhus | 0 |
| *Angiostrongylus cantonensis* infection | 8 | Salmonellosis | 0 |
| Cryptosporidiosis | 8 | Japanese encephalitis | 0 |
| *Streptococcus suis* type 2 infection | 8 | Cryptosporidiosis | 0 |
| Rabies | 7 | *Streptococcus suis* type 2 infection | 0 |
| Epidemic typhus | 7 | Rabies | -1 |
| Herpesvirus B infection | 7 | Epidemic typhus | -1 |
| Yellow fever | 7 | Herpesvirus B infection | -1 |
| Lassa fever | 7 | Yellow fever | -1 |
| Trichinosis | 7 | Lassa fever | -1 |
| Dengue fever | 5 | Trichinosis | -1 |
| Endemic typhus | 5 | Dengue fever | -3 |
| Lyme disease | 5 | Endemic typhus | -3 |
| Toxoplasmosis | 5 | Lyme disease | -3 |
| Blastocystosis | 5 | Toxoplasmosis | -3 |
| Zika viral infection | 4 | Blastocystosis | -3 |
| SFTS | 4 | Leptospirosis | -4 |
| Brucellosis | 4 | Brucellosis | -4 |
| *Paragonimus* infection | 4 | Bovine tuberculosis | -4 |
| Psittacosis | 4 | *Paragonimus* infection | -4 |
| Cat-scratch disease | 4 | Psittacosis | -4 |
| *Clonorchis sinensis* infection | 4 | Cat-scratch disease | -4 |
| SARS | 3 | *Clonorchis sinensis* infection | -4 |
| West Nile fever | 3 | Hantavirus syndrome | -4 |
| Rift Valley fever | 3 | Novel influenza A virus infection | -5 |
| MERS-CoV infection | 3 | Anthrax | -5 |
| Marburg viral hemorrhagic fever | 3 | Plague | -5 |
| New variant Creutzfeldt-Jakob disease | 3 | Ebola viral infection | -5 |
| Hendra viral infection | 3 | Tularemia | -5 |
| Nipah viral infection | 3 | Severe COVID-19 infection | -7 |

(*Continued*)

**Table 7.** (Continued)

| Sub-model A.1 | The score | Sub-model A.2 | The score |
|---|---|---|---|
| Chikungunya fever | 1 | Q fever | -7 |

[a] Models A.1 and A.2, were respectively constructed based on the different prioritization needs, namely "the characteristics of the disease itself and the ability to prepare medical resources" or "the need for stricter border controls and enhanced research on vaccine development or therapeutic drugs".

**Table 8. Univariate analysis of the evaluated variables in disease prioritization.**

| Factor | Zoonoses requiring 24-hour notification. (n = 23) | Zoonoses not requiring 24-hour reporting. (n = 23) | p-value |
|---|---|---|---|
| Case-fatality rate in humans over 5% | 20 (87%) | 13 (57%) | 0.02* |
| Case-fatality rate in animals over 5%[a] | 10 (83%) | 9 (60%) | 0.24 |
| The disease ever occurred in humans in a country, 2018–2020[b] | 9 (39%) | 13 (81%) | 0.01* |
| The disease ever occurred in animals in a country, 2018–2020[c] | 3 (100%) | 4 (67%) | 0.50 |
| The disease ever occurred human outbreaks in a country | 8 (35%) | 4 (17%) | 0.18 |
| The transmission mode of a disease | | | |
| Contact | 13 (57%) | 9 (39%) | 0.24 |
| Inhalation | 9 (39%) | 6 (26%) | 0.35 |
| Fecal-oral | 4 (17%) | 15 (65%) | 0.01* |
| Arthropod-borne | 9 (39%) | 5 (22%) | 0.2 |
| Human-to-human | 10 (44%) | 4 (17%) | 0.06* |
| The disease with therapeutic drugs | 12 (52%) | 19 (83%) | 0.03* |
| Reported with antimicrobial resistance | 5 (42%) | 8 (42%) | 1.00 |
| Available vaccine for prevention in humans | 10 (44%) | 4 (17%) | 0.06* |
| Available vaccine for prevention in animals | 7 (30%) | 9 (39%) | 0.54 |
| With the ability to infect economic animals | 11 (48%) | 19 (83%) | 0.01* |
| The quarantine required in imported animals | 3 (13%) | 7 (30%) | 0.15 |
| Type of the pathogen | | | |
| Bacteremia | 6 (26%) | 12 (52%) | 0.07* |
| Virus | 17 (74%) | 3 (13%) | 0.001* |
| Parasites | 0 (0%) | 5 (22%) | 0.02* |
| Others | 0 (0%) | 3 (13%) | 0.07 |
| Bioterrorism agents | 9 (39%) | 6 (26%) | 0.35 |
| Report to WOAH required | 5 (22%) | 7 (31%) | 0.50 |
| Report to WHO required | 10 (3%) | 0 (0%) | 0.001* |
| The mean risk score of a country associated with Taiwan | 12.09 | 16.83 | 0.09* |

*$p < 0.1$.

[a] A total of 27 diseases with available information (12 diseases in the group of notifiable zoonoses within 24 hours and 15 diseases in the group of Zoonoses not required to be reported within 24 hours).

[b] A total of 39 diseases with available information (23 diseases in the group of notifiable zoonoses within 24 hours and 16 diseases in the group of Zoonoses not required to be reported within 24 hours).

[c] A total of 8 diseases with available information (3 diseases in the group of notifiable zoonoses within 24 hours and 5 diseases in the group of Zoonoses not required to be reported within 24 hours).

Table 9. The comparison of the variables and their weights used in calculation of the prioritization scores in sub-models B.1 and B.2.

| Variables | Notifiable zoonoses within 24 hours (n = 23) | Zoonoses not required to be reported within 24 hours (n = 23) | Odds ratio | The weight value used for score calculation[a] | The weight impact in Model B.1[b] | The weight impact in Model B.2[b] |
|---|---|---|---|---|---|---|
| Case-fatality rate in humans over 5% | 20 (87%) | 13 (57%) | 4.07 | 4 | + | + |
| The disease with therapeutic drugs | 12 (52%) | 19 (83%) | 0.22 | 4 | + | - |
| Available vaccine for prevention in humans | 10 (44%) | 4 (17%) | 4.30 | 4 | + | - |
| With the ability to infect economic animals | 11 (48%) | 19 (83%) | 0.20 | 4 | + | + |

[a]The weight for each factor was determined based on the obtained odds ratio (OR) value: OR ≥ 4 or ≤ 0.25 received a weight of 4; OR values between 3–4 or 0.25–0.33 received a weight of 3; OR values between 2–3 or 0.33–0.5 received a weight of 2, and OR values between 1–2 or 0.5–1 received a weight of 1.

[b]Models B.1 and B.2, were respectively constructed based on the different prioritization needs, namely "the characteristics of the disease itself and the ability to prepare medical resources" or "the need for stricter border controls and enhanced research on vaccine development or therapeutic drugs".

factor "human case ever occurred in Taiwan, 2018 to 2020" was not included in the model due to difficulties in obtaining accurate information for the disease, especially for the controls that include most of the diseases not reported within 24 hours and might underestimate its importance.

According to the analysis by multiple logistic regression model, the four factors "case-fatality rate > 5% in humans", "the disease with therapeutic drugs", "available vaccine for prevention in humans", and "with the ability to infect economic animals" were assigned weights based on their odds ratio values (please refer to Table 9). The weights for the "the risk score of a country associated with Taiwan" and "multiple transmission routes" for each disease were listed in Table 4 and 5, respectively. Once the weights for each factor were determined, the final score for disease prioritization was calculated using the constructed model.

Model B.1 is constructed based on the consideration of "disease characteristics and the availability of medical resources" (Table 9). The formula for calculating each disease priority ranking score is as follows: Total score = 4 * (case-fatality rate > 5% in humans) + 4 * (the disease with therapeutic drugs) + 4 * (available vaccine for prevention in humans) + 4 * (with the ability to infect economic animals) + the weight score for countries with close exchanges + the weight score for transmission routes. The results show that the diseases with the highest ranking are anthrax, leptospirosis, novel influenza A viral infection, and bovine tuberculosis, all scoring 22 points. The diseases with the second-highest ranking are melioidosis, Q fever, and salmonellosis, all scoring 20 points. Listeriosis and *Streptococcus suis* type 2 infection rank third, both scoring 19 points. The scores and rankings of other diseases are listed in Table 10.

Model B.2 is constructed based on the consideration of "the need for stricter border controls and enhanced research on vaccines or therapeutic drugs for the disease." In comparison to Model B.1, the weights for "the disease with therapeutic drugs" and "available vaccine for prevention in humans" remain the same but subjectively assigned to negative values (Table 9). The formula for calculating the disease priority ranking score is as follows: Total score = 4 * (case-fatality rate > 5% in humans) - 4 * (the disease with therapeutic drugs) - 4 * (available vaccine for prevention in humans) + 4 * (with the ability to infect economic animals) + weight score for countries with close exchanges + weight score for transmission routes. The results show that the disease with the highest ranking is Zika viral infection, scoring 15 points. The diseases with the second-highest ranking are melioidosis, salmonellosis, Nipah viral infection,

**Table 10. Comparing prioritization results and disease scores from models B.1 and B.2[a].**

| Sub-model B.1 | The score | Sub-model B.2 | The score |
|---|---|---|---|
| Anthrax | 22 | Zika viral infection | 15 |
| Leptospirosis | 22 | Melioidosis | 12 |
| Novel influenza A virus infection | 22 | Salmonellosis | 12 |
| Bovine tuberculosis | 22 | Nipah viral infection | 12 |
| Melioidosis | 20 | West Nile fever | 12 |
| Q fever | 20 | SFTS | 12 |
| Salmonellosis | 20 | Rift Valley fever | 11 |
| Listeriosis | 19 | *Streptococcus suis* type 2 infection | 11 |
| *Streptococcus suis* type 2 infection | 19 | New variant Creutzfeldt-Jakob disease | 11 |
| Rabies | 18 | Listeriosis | 11 |
| Plague | 18 | Rabies | 10 |
| Enteropathogenic *Escherichia coli* | 18 | Enteropathogenic *Escherichia coli* | 10 |
| Japanese encephalitis | 18 | Japanese encephalitis | 10 |
| Tularemia | 18 | Trichinosis | 10 |
| Trichinosis | 18 | MERS-CoV infection | 9 |
| Scrub typhus | 17 | Cryptosporidiosis | 9 |
| Cryptosporidiosis | 17 | Scrub typhus | 9 |
| Hantavirus syndrome | 16 | SARS | 7 |
| Ebola viral infection | 16 | Marburg viral hemorrhagic fever | 7 |
| Zika viral infection | 15 | Toxoplasmosis | 7 |
| Severe COVID-19 infection | 15 | Brucellosis | 7 |
| Toxoplasmosis | 15 | Dengue fever | 6 |
| Brucellosis | 15 | Chikungunya fever | 6 |
| Dengue fever | 14 | Anthrax | 6 |
| *Paragonimus* infection | 14 | Leptospirosis | 6 |
| Blastocystosis | 14 | Novel influenza A virus infection | 6 |
| *Angiostrongylus cantonensis* infection | 13 | Bovine tuberculosis | 6 |
| Cat-scratch disease | 13 | *Paragonimus* infection | 6 |
| *Clonorchis sinensis* infection | 13 | Blastocystosis | 6 |
| West Nile fever | 12 | Hendra viral infection | 6 |
| SFTS | 12 | *Angiostrongylus cantonensis* infection | 5 |
| Lassa fever | 12 | Cat-scratch disease | 5 |
| Lyme disease | 12 | *Clonorchis sinensis* infection | 5 |
| Psittacosis | 12 | Lassa fever | 4 |
| Nipah viral infection | 12 | Lyme disease | 4 |
| Herpesvirus B infection | 11 | Q fever | 4 |
| Rift Valley fever | 11 | Psittacosis | 4 |
| New variant Creutzfeldt-Jakob disease | 11 | Herpesvirus B infection | 3 |
| Epidemic typhus | 10 | Plague | 2 |
| Yellow fever | 10 | Epidemic typhus | 2 |
| MERS-CoV infection | 9 | Yellow fever | 2 |
| Endemic typhus | 9 | Tularemia | 2 |
| SARS | 7 | Endemic typhus | 1 |
| Marburg viral hemorrhagic fever | 7 | Hantavirus syndrome | 0 |
| Chikungunya fever | 6 | Ebola viral infection | 0 |

*(Continued)*

**Table 10.** (Continued)

| Sub-model B.1 | The score | Sub-model B.2 | The score |
|---|---|---|---|
| Hendra viral infection | 6 | Severe COVID-19 infection | -1 |

[a]Models B.1 and B.2, were respectively constructed based on the different prioritization needs, namely "the characteristics of the disease itself and the ability to prepare medical resources" or "the need for stricter border controls and enhanced research on vaccine development or therapeutic drugs".

West Nile fever and SFTS, all scoring 12 points. Rift Valley fever, *Streptococcus suis* type 2 infection, new variant Creutzfeldt-Jakob disease and listeriosis rank third, all scoring 11 points. The scores and rankings of other diseases are listed in Table 10.

## 4. Discussion and conclusions

The construction of the model for disease prioritization requires a scientific and objective evaluation to include factors, as well as consideration of the relevant weight for the factor. This study showed several new directions for prioritization on scientific basis. Firstly, to address these research focuses, Model A utilizes a literature-based approach to establish a prioritization scoring model using commonly cited factors from the past studies. On the other hand, Model B adopts a more objective approach based on whether a disease needs to be reported within 24 hours to the government, aiming to identify important influencing factors and establish a prioritization scoring model. Secondly, regarding the assignment of weights for each factor, an objective method by logistic regression is employed and the results based on the range of odds ratio (OR) value was used as the reference to assign the weight. In this study, we also developed a scientific platform to determine the risk score of an associated country and the score regarding a disease with multiple transmission routes. Finally, we raised a new idea for two different purposes of disease prioritization, including "the characteristics of the disease and the availability of medical resources" or "the need for stricter border controls and enhanced research on vaccines or therapeutic drugs". A negative or positive value needs to further consider different prioritization goals and subjectively assigned to the weight of each factor in order to calculate the prioritization score for a disease. Therefore, this study not only provides zoonoses prioritization results for medical references, but also presents innovative research methods for studying disease prioritization.

The results of this study, compared with similar studies conducted in other countries, show differences regarding prioritization results of disease rankings. These differences may be attributed not only to the different approaches used for model construction but also to the uniqueness of each country's situation. These different also highlight the importance of this research, indicating that each country needs to consider not only the characteristics of the diseases but also the overall national context to study disease prioritization. It is not appropriate to directly adopt the ranking results developed in other countries without careful consideration.

Although there is no complete alignment between the disease prioritization orders of Taiwan and other countries, some diseases remain high priority in both Taiwan and other countries. For instance, the novel influenza A viral infection is considered a high-priority disease in the Netherlands, Vietnam, India, Uganda, and Colombia [9,13,21,28,56]. Anthrax is also regarded as a high-priority disease in India, Kenya, Uganda, and Australia [19,27,28,57]. Q fever is a high-priority disease in Italy and Australia [29,57]. West Nile fever is a high-priority disease in Burkina Faso and Canada [14,22]. These cases demonstrate that these zoonoses, due to their high case- fatality rates in humans, receive attention in multiple countries.

One of the most significant diseases that recently garnered international attention is monkeypox. On July 23, 2022, the World Health Organization (WHO) declared monkeypox a global public health emergency, as it started to spread in Europe and North America since May 2022. Monkeypox is classified into the West African strain and the Congo strain. The current epidemic is caused by a virus strain similar to the West African strain, with a fatality rate of 3.6% [58]. The disease mainly spreads through contact and inhalation, caused by a virus with no known infection in economic animals. Although there is no specific antiviral treatment for monkeypox, tecovirimat used to treat smallpox can be employed for treatment. The live attenuated vaccine can also confer protection two weeks after two doses [59]. If monkeypox is further assessed using the Model A.1 and A.2 frameworks, its scoring results are 9 and -7, respectively. Under Model B.1 and B.2, the scores are 14 and -2, respectively. Comparing these results with the ranking outcomes for zoonotic diseases in Tables 4 and 7, monkeypox ranks the third tied with Q fever and severe COVID-19 infection in A.1, and the eighth tied with toxoplasmosis and cryptosporidiosis in B.1. In Models A.2 and B.2, monkeypox ranks last tied with Q fever and severe COVID-19 infection in A.2, and the last one without any disease ties in B.2. Therefore, when considering the importance of monkeypox in the future, clear objectives should be set based on either "medical resource preparedness" or "stricter border controls and enhanced research on vaccines or therapeutic drugs" to ensure a scientifically and objectively founded consideration. This outcome also demonstrates the predictive and applicative nature of the models used in this research. Considering the overall needs of national epidemic prevention agencies, it is recommended to prioritize consideration based on the ranking results under the category of "characteristics of the disease and the availability of medical resources".

The priority order of diseases may change over time or with the emergence of new infectious diseases. Therefore, it is necessary to regularly re-assess the priority of diseases. However, no specific research has determined how often disease prioritization should be evaluated. The WHO's R&D Blueprint is a global strategic and preparedness plan designed to rapidly initiate research and development during major disease outbreaks. Its first disease prioritization was conducted in 2015 and subsequent reevaluations were done in 2017 and 2018 [60]. The European CDC recommends periodic reevaluation when disease drivers change or when new diseases emerge that could affect rankings [5]. Among countries that have previously conducted disease prioritization, Germany did so in 2008 and 2011 [7,10]. In our study, seven criteria were considered: case-fatality rate > 5% in humans, effective treatment available, existence of preventive measures for humans, historical occurrence in Taiwan, ability to infect economic animals, level of risk in closely interacting countries, and modes of disease transmission. While criteria related to infecting economic animals and disease transmission routes are less likely to change due to pathogen characteristics, advancements in disease monitoring and detection methods may provide new scientific evidence. For criteria related to the case-fatality rate, effective treatment availability, existence of preventive measures for humans, historical occurrence in a country, and level of risk in closely interacting countries, continuous research could also be changed according to new findings. Therefore, periodic reevaluation is necessary when changes in time and new scientific data emerge, to re-assess the results regarding disease prioritization.

In comparison to other methods for evaluating disease priority, particularly OHZDP, our models hold the advantage of assessing disease priority with less manpower and in a more time-efficient manner. The OHZDP process brings together representatives from human, animal, and environmental health sectors, along with other relevant partners, to prioritize the most concerning zoonotic diseases for multisectoral One Health collaboration in a country, region, or an area. While the workshop demands both time and financial resources, our

model, although requiring data collection, excels in evaluating diseases with reduced manpower and greater time efficiency.

This study had some limitations. The case group and control group consisted of a total of 46 diseases, so adding more independent variables to the model could lead to instability of the constructed model. Some diseases have wild animals as their hosts, resulting in limited data that could be obtained relevant to animal occurrence (only three diseases in the case group and five in the control group) and animal fatality rate (only 12 diseases with available information in the case group and 14 in the control group). Due to these limitations in animal data, the study primarily focused on the impact of diseases on humans and could not thoroughly evaluate the economic impact of certain diseases. Future research should address this limitation and conduct more in-depth studies to modify the model. Regarding closely interacting countries, it is also needs to concern countries that may have limited accessibility of monitoring data for diseases. Diseases not included in monitoring data were collected from PubMed, and diseases published in literature typically have special or severe characteristics. Additionally, this study used known zoonotic diseases to construct the model, but the global prevalence of emerging pathogens is constantly evolving. Therefore, future research should continually incorporate newly discovered pathogens for renew disease prioritization results.

## Author Contributions

**Conceptualization:** Chao-Chin Chang.

**Data curation:** Kung-Ching Wang.

**Formal analysis:** Kung-Ching Wang, Chao-Chin Chang.

**Methodology:** Chao-Chin Chang.

**Supervision:** Chia-Lin Chang, Sung-Hsi Wei, Chao-Chin Chang.

**Writing – original draft:** Kung-Ching Wang.

**Writing – review & editing:** Chia-Lin Chang, Chao-Chin Chang.

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
