## [Decision Letter · Decision Letter 0]

18 Jan 2024

PONE-D-23-32998The Study on Setting Priorities of Zoonotic Agents for Medical Preparedness and Allocation of Research ResourcesPLOS ONE

Dear Dr. Chang,

Thank you for submitting your manuscript to PLOS ONE. After careful consideration, we feel that it has merit but does not fully meet PLOS ONE’s publication criteria as it currently stands. Therefore, we invite you to submit a revised version of the manuscript that addresses the points raised during the review process.

We look forward to receiving your revised manuscript.

Kind regards,

Balbir B. Singh, Ph. D

Academic Editor

PLOS ONE

“The study was partially supported by the funding support (NSTC-111-2313-B-005-045-MY2) from National Science and Technology Council, Executive Yuan, Taiwan.”

5. In the online submission form you indicate that your data is not available for proprietary reasons and have provided a contact point for accessing this data. Please note that your current contact point is a co-author on this manuscript. According to our Data Policy, the contact point must not be an author on the manuscript and must be an institutional contact, ideally not an individual. Please revise your data statement to a non-author institutional point of contact, such as a data access or ethics committee, and send this to us via return email. Please also include contact information for the third party organization, and please include the full citation of where the data can be found.

Reviewers' comments:

Reviewer's Responses to Questions

**Comments to the Author**

1. Is the manuscript technically sound, and do the data support the conclusions?

Reviewer #1: Yes

Reviewer #2: Partly

2. Has the statistical analysis been performed appropriately and rigorously? 

Reviewer #1: Yes

Reviewer #2: No

3. Have the authors made all data underlying the findings in their manuscript fully available?

Reviewer #1: Yes

Reviewer #2: Yes

4. Is the manuscript presented in an intelligible fashion and written in standard English?

Reviewer #1: Yes

Reviewer #2: Yes

5. Review Comments to the Author

Reviewer #1: Very well written. Excellent methods and statistics. Topic very timely and relevant.

Minor edits:

3. Results

The results showed that the four most commonly used conditions, were “severity of the disease in humans”, “occurrence of the disease in the country”, and “curability in humans” (Table 2).

Should be: “showed that the three most commonly..” You discuss the 4th condition in the next paragraph an summarize the 4 conditions at the end of this paragraph.

Page 22: Moreover, as “pathogen type” is highly correlated with “human case fatality rate >5%” (e.g., viral infections), it was also excluded from the model to avoid r potential collinearity.

Remove the ‘r’

Reviewer #2: Review Report for the Article: Setting Priorities of Zoonotic Agents for Medical Preparedness and Allocation of Research Resources

The manuscript addresses the crucial topic of prioritizing zoonotic diseases, aiming to provide region-specific models and underscore the significance of adaptive preparedness. While commendable in its focus, the manuscript exhibits certain limitations, particularly in its methodology. Implementing the recommended enhancements will not only improve the clarity of the manuscript but also amplify its broader impact.

Introduction: The introduction lacks specific details on existing zoonoses prioritization tools and their inapplicability in the study region. A more thorough exploration of these tools and the reasons behind their inadequacy is necessary.

1. Page 3: “Once a zoonosis has been introduced into a country, factors such as urbanization and an aging population increase the likelihood of disease transmission and cause high case-fatality rates in humans.”: There can be many other factors, especially pathogen related, for deciding the likelihood of disease transmission and high case-fatality rates in humans. So, please rewrite it.

2. Page 3: “Common methods used for this purpose include the Hirsch index (h-index), Delphi technique, multicriteria decision analysis (MCDA), questionnaires, and the OHZDP tool, each with its advantages and disadvantages”: As I understand only CDC’s OHZDP tool is meant for zoonoses prioritization. Please correct it.

3. Expand the term “OHZDP tool”

Material and methods:

General comments: The authors should have followed PRISMA guidelines for literature review to keep the extracted data more rigorous. Overall, the methodology can be improved by explaining the steps clearly or by providing detailed flow chart.

1. Page 5: “Through careful review of the 25 relevant articles, frequency of the 17 criteria were then summarized, and the top 3 criteria were identified and further used for zoonosis prioritization in this study.”: I would advice to provide a table about the methods used in these selected 17 studies.

2. “Table 1. The list of zoonoses used for disease prioritization in this study.”: Already this classification for notifying the zoonoses is there in the region. How can the authors claim that they have drafted more important classification criteria?

3. Page 7: “To identify whether the outbreak has been ever occurred in Taiwan and any arthropod vector responsible for the transmission, we used the disease name and "Taiwan" as keywords to search for relevant literature in PubMed and the epidemic report from Taiwan CDC”: I don’t think that this is correct search strategy to find vector/arthropod related infectious diseases. Please relook it.

4. Page 9: “and corresponding scores were assigned based on the calculated odds ratio (OR) values”: I couldn’t understand that how the authors calculated Odds ratio for the particular disease?

5. Page 9: “Based on the OR values after such statistical analysis, a score of 4 was assigned for foodborne or person-to-person transmission, 3 for contact or vector-borne transmission, and 2 for airborne transmission.”: Here I would argue that airborne transmission should have more weightage. Please explain the logic behind these scoring

Results

1. Table 2: I believe that mode of transmission is important factor to include, although there might be less literature on this, but on this basis, it is not correct to ignore this factor.

2. Page 14: “The score presented the sum of the risk from various countries and was calculated based on the disease ever occurred in China, Japan and Hong Kong (score=4), in South Korea and USA (score=3), in Thailand, Viet Nam and Singapore (score=2) and in Malaysia, Philippines and Indonesia (score=1)”. The basis of the scoring needs to be explained.

3. Page 18: I think there might be many other potential factors which might affect the output of model A1 and A2 (e.g., contagiousness, availability of diagnostics, DALYs, bioterrorism potential etc.)

Discussion: Overall, the discussion need improvement. The authors can be discussed about the rationale behind the prioritized zoonoses. There is need to discuss the merits and demerits of the developed model with the existing model, especially with CDC’s OHZDP.

6. PLOS authors have the option to publish the peer review history of their article (what does this mean?). If published, this will include your full peer review and any attached files.

Reviewer #1: No

Reviewer #2: **Yes: **Pankaj Dhaka

---

## [Author Response · Author response to Decision Letter 0]

7 Feb 2024

PLOS ONE

Manuscript Number: PONE-D-23-32998

The Study on Setting Priorities of Zoonotic Agents for Medical Preparedness and Allocation of Research Resources 

Review Comments to the Author

Reviewer #1: 

Very well written. Excellent methods and statistics. Topic very timely and relevant.

Minor edits:

3. Results

The results showed that the four most commonly used conditions, were “severity of the disease in humans”, “occurrence of the disease in the country”, and “curability in humans” (Table 2).

Should be: “showed that the three most commonly..” You discuss the 4th condition in the next paragraph an summarize the 4 conditions at the end of this paragraph.

The authors’ reply: Thank you for your kind reminder. We have replaced the word ‘four’ with ‘three’ on page 11. 

Page 22: Moreover, as “pathogen type” is highly correlated with “human case fatality rate >5%” (e.g., viral infections), it was also excluded from the model to avoid r potential collinearity.

Remove the ‘r’

The authors’ reply: Thank you for your kind reminder. We have deleted the ‘r’ on page 23.

 

Reviewer #2: 

Review Report for the Article: Setting Priorities of Zoonotic Agents for Medical Preparedness and Allocation of Research Resources The manuscript addresses the crucial topic of prioritizing zoonotic diseases, aiming to provide region-specific models and underscore the significance of adaptive preparedness. While commendable in its focus, the manuscript exhibits certain limitations, particularly in its methodology. Implementing the recommended enhancements will not only improve the clarity of the manuscript but also amplify its broader impact.

The authors’ reply: Thanks for your kind comments. We have revised the manuscript according to the following comments raised by the reviewer. The authors sincerely believe that the revised manuscript will make the research more meaningful, contribute innovative methodology and offer objective platform to this specific but important topic, as the reviewer expected. 

Introduction: The introduction lacks specific details on existing zoonoses prioritization tools and their inapplicability in the study region. A more thorough exploration of these tools and the reasons behind their inadequacy is necessary.

1. Page 3: “Once a zoonosis has been introduced into a country, factors such as urbanization and an aging population increase the likelihood of disease transmission and cause high case-fatality rates in humans.”: There can be many other factors, especially pathogen related, for deciding the likelihood of disease transmission and high case-fatality rates in humans. So, please rewrite it.

The authors’ reply: Thank you for your suggestion. We have made additional modifications according to the reviewer’s comment accordingly: “Moreover, several factors related to the characteristics of the pathogen, including its mode of transmission, influence the magnitude of an epidemic of a zoonosis. The availability of therapeutic agents or vaccines to prevent viral or bacterial diseases also determines the epidemic scale of a zoonosis once occurred.”. Please refer to the information on page 3.

2. Page 3: “Common methods used for this purpose include the Hirsch index (h-index), Delphi technique, multicriteria decision analysis (MCDA), questionnaires, and the OHZDP tool, each with its advantages and disadvantages”: As I understand only CDC’s OHZDP tool is meant for zoonoses prioritization. Please correct it.

The authors’ reply: Thanks for the information. In fact, OHZDP has been recommended by CDC in the US as a tool for zoonoses prioritization. However, other methods have been applied in different countries after literature review. To more clarify the information according to the reviewer’s comment, we revised the sentence as “Up to date, the methodology of One Health Zoonotic Disease Prioritization (OHZDP) has been recommended by CDC in the US as a tool for zoonoses prioritization, and various methods used for this purpose in different countries include the Hirsch index (h-index), Delphi technique, multi-criteria decision analysis (MCDA), and questionnaires; each method is with its advantages and disadvantages [4]”. Please refer to the modification on pages 3-4.

3. Expand the term “OHZDP tool”

The authors’ reply: Thank you for your kind reminder. We have expanded OHZDP tool as “One Health Zoonotic Disease Prioritization” on page 3.

Material and methods:

General comments: The authors should have followed PRISMA guidelines for literature review to keep the extracted data more rigorous. Overall, the methodology can be improved by explaining the steps clearly or by providing detailed flow chart.

The authors’ reply: Many thanks for the reviewer’s suggestion. After consideration, we think that the main focus of the manuscript is to establish a platform for prioritization of zoonoses, and is not for meta-analysis. Therefore, for the part of literature review and not to make the manuscript too long, we decided to briefly describe the methodology in the materials and methods (please refer to the information on pages 5-7). We will apply PRISMA guidelines for reporting systematic reviews and meta-analysis in our future study.

1. Page 5: “Through careful review of the 25 relevant articles, frequency of the 17 criteria were then summarized, and the top 3 criteria were identified and further used for zoonosis prioritization in this study.”: I would advice to provide a table about the methods used in these selected 17 studies.

The authors’ reply: Thank you for your suggestion. We have provided a table listing the prioritization methods used in the relevant 25 studies on pages 6-7.

2. “Table 1. The list of zoonoses used for disease prioritization in this study.”: Already this classification for notifying the zoonoses is there in the region. How can the authors claim that they have drafted more important classification criteria?

The authors’ reply: Sorry for the confusion. Here, we would like to clarify that the list was established through expert meetings to merely determine whether the disease needs to be reported within 24 hours (i.e., two groups) or not. However, those diseases have not been yet determined their prioritized ranks. Consequently, it is of major importance that this study was conducted, and used the list of the diseases to establish criteria and a platform for aiding in prioritizing zoonoses. 

3. Page 7: “To identify whether the outbreak has been ever occurred in Taiwan and any arthropod vector responsible for the transmission, we used the disease name and "Taiwan" as keywords to search for relevant literature in PubMed and the epidemic report from Taiwan CDC”: I don’t think that this is correct search strategy to find vector/arthropod related infectious diseases. Please relook it.

The authors’ reply: Thank you for your suggestion. While there are potentially other methods, such as entomological investigations or large-scale epidemiological studies for identifying vector/arthropod-related infectious diseases, it is not feasible within the scope of our study and could be conducted according to the results of disease prioritization in the future. We acknowledge that our chosen method is not perfect; nevertheless, we still consider that it remains a practical way to use the available data through checking of the widely-used public reporting systems for identification of occurrence of vector-borne diseases in Taiwan. 

4. Page 9: “and corresponding scores were assigned based on the calculated odds ratio (OR) values”: I couldn’t understand that how the authors calculated Odds ratio for the particular disease?

The authors’ reply: The calculated odds ratio (OR) was obtained through the analysis of a binary logistic regression model by a case-control study design, as described in the materials and methods (please see pages 4-5 for more information). Briefly, a binary logistic regression model is used for constructing a multivariate model which include most influential factors associated with prioritization of zoonoses identified in our study. The factors influencing the prioritization of zoonoses in the binary model were determined through a case-control study, where diseases requiring 24-hour notification constituted the case group and those not requiring 24-hour notification constituted the control group for statistical analysis. 

Two models are created: Model A is constructed based on four commonly used criteria extracted from relevant research literature that studied prioritization of infectious diseases; Model B initially conducts a univariate analysis of factors that may influence the need for notification within 24 hours. In the Model B analysis, if the p-value from the single-factor analysis is less than 0.1, the factor is incorporated into constructing the multivariate model. After the model has been constructed, on the basis of the concept of a binary logistic regression, the coefficient of each influential factor can be obtained and the natural constant “e” to the power of the coefficient can be calculated to get the OR value. 

The weight assigned to each factor is determined based on the obtained odds ratio (OR) value: factors with OR ≥ 4 or ≤ 0.25 are assigned a weight of 4; OR values falling between 3-4 or 0.25-0.33 receive a weight of 3; OR values ranging from 2-3 or 0.33-0.5 receive a weight of 2, and OR values between 1-2 or 0.5-1 are assigned a weight of 1. Once the weight for each factor is established, the total score for each disease is calculated for prioritization using the overall equation formulated for this purpose.

5. Page 9: “Based on the OR values after such statistical analysis, a score of 4 was assigned for foodborne or person-to-person transmission, 3 for contact or vector-borne transmission, and 2 for airborne transmission.”: Here I would argue that airborne transmission should have more weightage. Please explain the logic behind these scoring

The authors’ reply: The comments from the reviewer are well-received. In this study, this study offers an objective way to suggest that the determination of the weighted score of a factor should ideally be based on scientific analysis and a more objective approach, rather than relying solely on professional judgement. This again emphasizes the importance of this study. Furthermore, we used a binary logistic regression model to construct a multivariate model to assess the transmission route score, recognizing that many significant zoonoses may involve multiple transmission routes, with some routes having equal transmission efficiency. The dependent variable in this binary model is derived from the concept of a case-control study; diseases necessitating notification within 24 hours are designated as the case group, while those not requiring notification within 24 hours serve as the control group. The independent variables related to mode of transmission include contact transmission, airborne transmission, foodborne transmission, vector-borne transmission, and person-to-person transmission. Finally, binary logistic regression analysis was used to assign objective scores to each mode of transmission based on the odds ratio (OR) values obtained. This approach allows for a rigorous and data-driven evaluation of the transmission pathways, thereby increasing the stability and reliability of the results.

Results

1. Table 2: I believe that mode of transmission is important factor to include, although there might be less literature on this, but on this basis, it is not correct to ignore this factor.

The authors’ reply: Thank you for your suggestion. We totally agree with this idea. Therefore, in our study, we have different approaches for model construction for prioritization of zoonoses. In Model A, we incorporated the four most commonly used conditions through literature summarization to construct the model. Notably, the mode of transmission was ranked the eighth factor used for disease prioritization after careful summarization of the 25 relevant articles. Therefore, we did not include the mode of transmission in Model A for prioritization of zoonoses. In Model B, although the mode of transmission was initially selected for constructing the multivariate model, it was finally excluded due to the judgement by statistical significance, possibly due to collinearity with other factors. Overall, we consider that although the mode of transmission itself is conceptually with importance for disease prioritization, it could be explained by the other more influential factors in the model for prioritization of zoonoses. The final conclusion is that the best objective and scientific model for disease prioritization may not need to include many factors, but could be achieved by a simplified model including most influential factors for practical use.

2. Page 14: “The score presented the sum of the risk from various countries and was calculated based on the disease ever occurred in China, Japan and Hong Kong (score=4), in South Korea and USA (score=3), in Thailand, Viet Nam and Singapore (score=2) and in Malaysia, Philippines and Indonesia (score=1)”. The basis of the scoring needs to be explained.

The authors’ reply: Thank you for your suggestion. We have explained it on page 9-10. The sum of the risk of countries was conducted based on data from the Tourism Statistics Database of the Tourism Bureau of the Ministry of Transportation. The data collected spans from 2010 to 2020 and includes the number of visitors to Taiwan based on their country of residence and the number of Taiwanese nationals traveling abroad, categorized by the destination country. After merging the data, a total of 22 countries with complete information were identified, with the top 11 countries selected as those closely linked to Taiwan. The total was then calculated by summing the numbers of incoming tourists, outgoing Taiwanese travelers and migrant workers. Finally, these figures were ranked in descending order based on the total number of individuals, and the risk scores were assigned to four levels according to the quartiles of the total population.

3. Page 18: I think there might be many other potential factors which might affect the output of model A1 and A2 (e.g., contagiousness, availability of diagnostics, DALYs, bioterrorism potential etc.)

The authors’ reply: Thank you for your suggestion. We agreed that other potential factors might affect the output of models A1 and A2. However, based on the criteria described in the materials and methods regarding how to choose a factor in the model in our study, therankings of the factors suggested by the reviewer among the 25 relevant articles on disease prioritization studies were insufficient to merit inclusion in model A.

Discussion: Overall, the discussion need improvement. The authors can be discussed about the rationale behind the prioritized zoonoses. There is need to discuss the merits and demerits of the developed model with the existing model, especially with CDC’s OHZDP.

The authors’ reply: Thank you for your suggestion; we have added this paragraph for further discussion on pages 32-33. “In comparison to other methods for evaluating disease priority, particularly OHZDP, our models hold the advantage of assessing disease priority with less manpower and in a more time-efficient manner. The OHZDP process brings together representatives from human, animal, and environmental health sectors, along with other relevant partners, to prioritize the most concerning zoonotic diseases for multisectoral One Health collaboration in a country, region, or other area. While the workshop demands both time and financial resources, our model, although requiring data collection, excels in evaluating diseases with reduced manpower and greater time efficiency.”

---

## [Editor Report · Decision Letter 1]

13 Feb 2024

The Study on Setting Priorities of Zoonotic Agents for Medical Preparedness and Allocation of Research Resources

PONE-D-23-32998R1

Dear Dr. Chang,

We’re pleased to inform you that your manuscript has been judged scientifically suitable for publication and will be formally accepted for publication once it meets all outstanding technical requirements.

Kind regards,

Balbir B. Singh, Ph. D

Academic Editor

PLOS ONE